# Cutaneous Manifestations in Biological-Treated Inflammatory Bowel Disease Patients: A Narrative Review

**DOI:** 10.3390/jcm10051040

**Published:** 2021-03-03

**Authors:** Jo L. W. Lambert, Sofie De Schepper, Reinhart Speeckaert

**Affiliations:** Department of Dermatology, Ghent University Hospital, B-9000 Gent, Belgium; sofie.deschepper@uzgent.be (S.D.S.); reinhart.speeckaert@uzgent.be (R.S.)

**Keywords:** inflammatory bowel disease, skin manifestations, tumor necrosis factor-alpha, biological therapy

## Abstract

The biologic era has greatly improved the treatment of Crohn’s disease and ulcerative colitis. Biologics can however induce a wide variety of skin eruptions, especially those targeting the TNF-α and Th17 pathway. These include infusion reactions, eczema, psoriasis, lupus, alopecia areata, vitiligo, lichenoid reactions, granulomatous disorders, vasculitis, skin cancer, and cutaneous infections. It is important to recognize these conditions as treatment-induced adverse reactions and adapt the treatment strategy accordingly. Some conditions can be treated topically while others require cessation or switch of the biological therapy. TNF-α antagonists have the highest rate adverse skin eruptions followed by ustekinumab and anti-integrin receptor blockers. In this review, we provide an overview of the most common skin eruptions which can be encountered in clinical practice when treating IBD (Inflammatory bowel disease) patients and propose a therapeutic approach for each condition.

## 1. Introduction

Crohn’s disease (CD) and ulcerative colitis (UC) sometimes present with extra-intestinal cutaneous manifestations, occurring in up to 55% (CD) and 35% (UC), respectively (EIMs) [1]. They can even appear before inflammatory bowel disease (IBD) diagnosis is clear, and should prompt any doctor to screen for IBD. These are usually classified according to their pathophysiological association with the underlying intestinal disease into four categories: 1/specific (the continuous mucocutaneous and metastatic non-caseating granulomas), 2/reactive (erythema nodosum, pyoderma gangrenosum, Sweet syndrome), 3/associated (psoriasis, hidradenitis suppurativa, vitiligo, phlebitis, erythema multiforme, urticaria, lichen planus, secondary amyloidosis, and various autoimmune blistering disorders), and 4/treatment induced-manifestations. The latter form the topic of this review and are hence summarized below [2,3,4,5].

Biological drugs have a very important position in the treatment of patients with inflammatory chronic conditions, such as inflammatory bowel disease (IBD). Ulcerative colitis (UC) and Crohn’s disease (CD) are the two major types of inflammatory bowel disease (IBD). Currently approved biologics are targeting tumor necrosis factor (TNF)-α (infliximab, adalimumab, certolizumab, golimumab), interleukin (IL)12/23 (ustekinumab), and integrin receptors (natalizumab and vedolizumab).

Despite the proven efficacy of biological drugs for inflammatory bowel disease, these therapies do carry some risk. The occurrence of skin side effects during therapy can hamper drug survival in patients, and can therefore increase morbidity. Between 20–25% of anti-tumor necrosis factor-alpha (anti-TNF-α)-treated patients have skin side effects in the course of their treatment which have been reported both in adults and children [6,7]. A causal relation is assumed through the correlation between the start of therapy and onset of skin eruption, through the fact that the eruption disappears after cessation of the therapy and through an ultimate rechallenging test causing the same eruption [8,9,10]. Diagnosis and correct management are warranted as they considerably influence morbidity and even mortality rates of the IBD patient.

In this narrative review, we take you on a tour, describing the most common skin problems under biological treatment in IBD patients. We describe a series of skin side effects, classifying them from ‘very frequent to rare’ (Table 1). We also propose how to manage those side effects. Furthermore, we put these events into a diagnostic and pathophysiologic frame, possibly leading to a set of predictive factors for these particular types of side effects. Pubmed, Embase, and Google Scholar were searched for articles mentioning cutaneous adverse events in IBD patients on biologics. The used search terms were ‘inflammatory bowel disease’, ‘tumor necrosis factor’, ‘Crohn’, ‘ulcerative colitis’, ‘TNF’, ‘adalimumab’, ‘certolizumab’, ‘infliximab’, ‘natalizumab’, ‘ustekinumab’, and ‘vedolizumab’. All types of articles were included. The exclusion criteria were articles without full-text availability or articles not written in English.

## 2. Pathogenetic Insights

Adverse events (AE) on biologics can be divided into 5 main categories: type α, type β, type y, type δ, and type ε [19]. Type α consists of a high cytokine and cytokine release syndrome, which involves side effects directly related to elevated concentrations of cytokines in the circulation. This does not apply to biologics used for IBD. Type β indicates a hypersensitivity response. This can be an acute allergic response (<30 min) induced by anti-IgE antibodies, clinically visible by a local wheal reaction at the injection site, or in more severe cases by urticaria of anaphylaxis. Hypersensitivity reactions can also be delayed (>6 h) if they are IgG, complement or T-cell mediated. Anti-drug-IgG antibodies occur frequently (up to 68% in case of infliximab) and can be neutralizing or non-neutralizing [20,21]. This can lead to the inactivation of the biologic or shortened half-life due to increased clearance of the immune complexes from the circulation. These immune complexes can also activate the complement system and activation of neutrophils via Fc-IgG resulting in serum sickness, nephritis, and/or vasculitis. Type y involves most of the ‘paradoxical’ skin eruption on biologics. They can be caused by changes in the immune or cytokine profile leading to autoimmunity or allergic disorders. Biologics targeting TNF-α can induce a wide range of immunological responses ranging from IFN-y dominant responses such as vitiligo and alopecia areata (Th1), Th17-mediated skin disorders such as psoriasis and palmoplantar pustulosis but also Th2-induced disorders such as eczema. The pathogenesis has not been fully elucidated although type I interferons (IFN-α) and plasmacytoid dendritic cells seem implicated in several TNF-induced skin eruptions including paradoxical psoriasis, psoriasiform dermatitis, and lupus [22]. Recently, more emphasis has been placed on the plasticity of the Th17 pathway. Th17 cells retain stem cell-like properties and can acquire Th1, Th2, or Treg properties. Drugs targeting this pathway (e.g., TNF-blockers, IL-23 blockers) can therefore skew Th17 differentiation into a Th1, Th2, or Treg activities depending on the surrounding immune environment. Interestingly, another important aspect is the formation of neutrophil extracellular traps (NETs). NETs are webs made with antimicrobial peptides and DNA which are used by neutrophils to catch extracellular pathogens. However, these mechanisms expose antigenic molecules (e.g., LL-37: the key autoantigen in psoriasis), histones, and DNA in the extracellular environment. This might explain the high rates of ANA and anti-dsDNA antibodies found in patients receiving TNF-blockers. NETs are activated by immune complexes and antidrug/anti-TNF complexes have been confirmed in the circulation of patients, particularly with infliximab [23]. NETs activate plasmacytoid dendritic cells which produce high levels of IFN-α [24]. Additionally, TNF-α inhibits IFN-α by reducing the development of plasmacytoid dendritic cells and inhibiting the release of IFN-α by virally stimulated immature plasmacytoid dendritic cells [25]. Finally, TNF-α also induces Treg activity. Its loss during biologic treatment could also account for clinical important inflammatory skin disorders. Type δ involves the effects of targeting the same antigen which is also present on non-pathogenic cells or to a similar structured protein or peptide. This category seems less important in the biologics used in IBD. An example of nonimmunologic side effects (type ε) is the aggravation of heart failure by neutralizing TNF-α.

## 3. Infusion and Injection-Site Reactions

Acute infusion reactions are seen with infliximab during the first 24 h after treatment. Patients show erythema, flushing, pruritus, and urticaria, and rarely evolve to an anaphylactic state. Delayed infusion reactions occur between 24 h and 14 days after infusion and present as urticaria vasculitis, and serum sickness. The incidence of infusion reactions to infliximab is reported to be around 6.1% involving 9.7% of patients. Severe acute reactions concerned only 1% of infusions. Mild reactions were characterized by hyperemia, palpitations, diaphoresis, headache, dizziness, and nausea. Moderate reactions were hypo-/hypertension, hyperemia, chest discomfort, shortness of breath, elevated temperature, palpitations, and urticaria. In severe reactions, significant hypertension, elevated temperature with rigors, hyperemia, chest discomfort, significant shortness of breath, and stridor were observed [11]. A retrospective study in an alternate setting (home of ambulatory infusion suite) found that 2% of all infusions of infliximab in IBD patients (both CD and UC) leads to infusion reactions affecting 7.8% of patients. Most reactions were mild to moderate and could be managed with rate adjustments and/or therapy. Emergency room admission was necessary for 0.1% of all infusions, and incompleted infusions due to reactions were observed in 0.3% of all infliximab infusions [26]. There is a bigger risk for infusion reactions (2–3x) when longer time intervals between the infusions are used because this triggers the formation of anti-drug antibodies against infliximab (IgE–IgG). In most cases, acute infusion reactions are not IgE-mediated, as successful treatment with reduced infusion rates is often possible. Moreover, wheezing—a key symptom of an allergic hypersensitivity response—is rare, and normal tryptase levels are detected [11]. Acute infusion reactions are often not immunological but related to the administration rate. Delayed reactions to infliximab are in general not linked to a type IV hypersensitivity response. They can be serum sickness-like and caused by anti-drug antibodies or type III hypersensitivity. Other possible signs are lupus-like symptoms, viral syndrome, IBD flare, and non-specific symptoms [11].

Injection site reactions are accompanied by erythema, itch, and swelling around the site of subcutaneous injection. This reaction is often mild and occurs 1–24 h after injection, peaking at 48 h. Although inducing pain and stress, the clinical relevance on the efficacy of the biologic is not obvious. We also see that this most often happens during the first months of therapy and lasts 3–5 days [27]. Due to the administration route, infliximab has less local reactions compared to adalimumab (5–20%), golimumab (6%), and certolizumab pegol (4.3–9.6%) [28,29]. Similarly, injection site reactions develop more frequently when vedolizumab is administered subcutaneously (10.4%) versus intravenously (1.9%) [30]. This side effect is rare for ustekinumab (1–2%) [29].

### Management

Acute infusion reactions due to infliximab can be managed by decreasing the infusion rate, antihistaminics (diphenhydramine), non-steroidal anti-inflammatory drugs (acetaminophen), and steroids. Prophylaxis protocols (e.g., pretreating with diphenhydramine and/or acetaminophen or steroids) have been proposed to prevent the recurrence of infusion reactions. Nonetheless, a meta-analysis has shown that the risk of acute infusion reactions seems not to decrease with pretreatment [31]. As such, in case of severe reactions, the risk versus benefit and alternative treatment options should be carefully considered [11].

Injection site reactions rarely require therapy [32]. The injection sites should be rotated and placed in non-bruised healthy skin. Most biologics should be left rested at room temperature for 30–45 min before injection. A cold compression can be applied afterward. Painkillers, topical steroids, or oral antihistaminics can be helpful. As autoinjectors always penetrate the skin at the same level, they are often preferred over prefilled syringes [29]. Injection site reactions are less when adalimumab is combined with methotrexate or cyclosporin [28].

## 4. Eczema

Eczema as an adverse effect of anti-TNF-α therapy may occur in approximately 5–20% of patients with inflammatory bowel disease (both CD and UC). Personal history of atopy appears to increase this risk [33]. Patients taking anti-TNF-α agents for prolonged periods may develop dry skin, especially in older patients and in the winter period [34] (Figure 1a). Dry skin may be a precursor sign of eczema and should be treated accordingly. Development of stasis dermatitis in patients with venous insufficiency of the legs has also been described with anti-TNF-α therapy [35] (Figure 1b).

A remarkable characteristic anti-TNF-α induced overlap syndrome termed ‘psoriasiform eczema’ or ‘psoriasiform dermatitis’ should be emphasized as it represents the most common skin eruption. It is important to recognize this disorder as a separate entity that is different from eczema and psoriasis. In these patients, both symptoms typical for (atopic) dermatitis (xerosis, pruritus, bacterial superinfection) and psoriasis (thick white scales, orange-red color) are present. Similarly, histological examination reveals aspects of both disorders. Bacterial superinfection, which is rarely seen in psoriasis, is remarkably frequent in anti-TNF-α-induced psoriasiform eczema [34].

Immunologic analysis of anti-TNF-α induced eczema and psoriasis detected elevated IL-17A, IL-23, and IFN-γ serum levels in CD patients with skin lesions. No significant difference could be found between the type of skin lesions illustrating their common pathogenesis.

Anti-TNF-α biologics strongly decrease the expression of IL-17 related cytokines. Reciprocal regulation between Th2 and Th17 pathways has been demonstrated. The Th2 cytokine IL-13, one of the major cytokines associated with eczema represses the expression of Th17 genes. Similarly, IL-17A reduces the transcription of Th2 genes. Although these experiments have been conducted on bronchial epithelial cells, they shed light on the counter-regulating signaling between both pathways and why a Th2 response can develop when blocking the Th17 pathway (AE type y, autoimmunity by immunomodulation) [36].

### Management

Despite the frequent occurrence of eczema in IBD patients treated with anti-TNF-α, no clear recommendations have been made on its management. Similar observations have been detected in psoriasis patients converting to an atopic eczema phenotype. In these patients, the anti-TNF biologic is stopped in the majority of cases, although conservative management (e.g., topical steroids,) can also be successful and considered as a first option [37]. General measures such as avoiding shower gel, using shower/bath oil and regularly applying emollients are recommended. In the case of superinfection, topical or systemic antibiotics (e.g., flucloxacillin) can be considered [34]. Switching to ustekinumab or anti-integrin receptor treatment can be necessary in recalcitrant cases.

## 5. Cutaneous Infections

Bacterial skin infections such as cellulitis and erysipelas occur with a prevalence of 0.3 to 3% (AE type γ, infections by immunosuppression). Viral infections that are most often seen are Herpes zoster and Varicella zoster in 1 to 5% of treated patients. Herpes simplex occurs in 0.7–1.7%. Fungal infections mainly present as tinea pedis in 0.9–6.9% of cases. Cutaneous infections with atypical Mycobacteria should not be overlooked and present as chronic papulonodular skin lesions, often crusty, and in a sporotrichoid (i.e., the appearance of subcutaneous nodules that progress along dermal and lymphatic vessels) pattern [38]. The development of soft skin tissue infections is linked to concomitant steroid use, especially in case of anti-TNF-α [39]. Other risk factors to get contracted with these micro-organisms are immunosuppressant combination therapy, malnutrition, and associated comorbidities such as diabetes. Ustekinumab is generally associated with lower rates of skin infections compared to anti-TNFs, while anti-integrin receptors carry no increased risk [39] Discontinuation of biologics before surgery is not substantiated by high evidence as the risk of post-operative soft skin tissue infections appear to be similar with or without biologics. Additionally, a temporary stop may lead to disease flares [39]. A nationwide Danish cohort study did not show increased serious infections in pedatric patients with IBD (CD: *n* = 432; UC or unclassified: *n* = 186) [40].

### Management

Given the increased rate of Varicella zoster infection in patients receiving anti-TNF treatment, especially combined with other immunosuppressants, varicella vaccination is recommended. Treatment of Varicella infection depends on the underlying immunosuppressive therapy, the severity of the disease, and if it concerns a reactivation or generalized clinical presentation [41]. In patients receiving both anti-TNF-α and other immunosuppressive agents, early recognition and appropriate treatment of skin tissue infections is advised as the risk for a more serious course is increased. The first 3 months after initiation seem of particular importance in IBD patients treated with anti-TNF-α [42].

## 6. Psoriasiform Reactions

In a 14-year retrospective study with 583 anti-TNF-α treated IBD patients, 20.5% showed dermatological complications. Of these, 10.1% were psoriasiform lesions (=type y, ‘paradoxal’ adverse event), which was predominantly found in patients with CD (CD: 10.8%; UC: 6.8%). Risk factors that were noted were: younger age, having CD, smoking, and higher dosing [43,44]. It often concerns de novo psoriasis lesions (infliximab-adalimumab) but sometimes it represents an exacerbation of pre-existing psoriasis (etanercept). As outlined in the section on eczema, most patients display an overlap between psoriasis and eczema which has been reported to be 10 times more frequent than true psoriasis in IBD patients receiving anti-TNF-α [45]. Sharp borders of the lesion, silvery-white scales, absence of superinfection, limited itch, and slow response to topical treatment all favor the diagnosis of psoriasis instead of psoriasiform dermatitis [34].

A disproportionate percentage (+/−40%) of pustular lesions on hands and feet is seen in comparison to the proportion of palmoplantar pustulosis (1.7%) detected in psoriasis patients [46]. Palmoplantar pustulosis presents with sterile pustules on the hands and feet on an erythematous background with scales (Figure 1c). It preferably affects non-weight-bearing areas. Palmoplantar pustulosis is often challenging to treat as the response to treatment is often moderate.

### Management

Dermatological treatment of psoriasis allows continuing anti-TNF-α in half of them. Local steroids, keratolytics, and vitamin D analogs help in case of limited lesions. In more resistant and extensive lesions, phototherapy or methotrexate, acitretin, and cyclosporine are used. Combination therapy of anti-TNF-α with other immunosuppressants is linked with a reduced risk of psoriasis [44]. Most of TNF-α induced skin lesions resolve after switch to ustekinumab. Ustekinumab has been proposed as the drug of choice in IBD patients with recalcitrant skin lesions [47]. Good responses have also been seen after switching to vedolizumab [48].

## 7. Cutaneous Malignancies

Melanoma and non-melanoma skin cancer (NMSC) risks are increased in IBD. Anderson et al. looked at a retrospective cohort of 2127 IBD patients (CD: 63.1%; UC: 36.9%), and the resulting incidence of non-melanoma skin cancer and melanoma was 35.4/10,000 (95% CI: 23.3–51.5) and 6.56/10,000 (95% CI: 2.1–15.3), respectively [49]. There appears to be an intrinsic risk of melanoma in IBD which is not explained by immunosuppressive treatments [50]. However, biological treatment of these patients may further increase the risk of developing these skin cancers, although the risk is probably higher with photosensitizing immunosuppressants such as thiopurines [49]. Anti-TNF-α is reported to lead to a 1.5–4-fold increased risk for melanoma in IBD patients (AE type γ, cancer through immunosuppression) [13,14,15]. Risk factors to develop these skin cancers are: solar exposition, skin phototype, and family history. No increased risk of NMSC has been shown for ustekinumab and the anti-integrin receptors antibodies vedolizumab and natalizumab [51,52].

### Management

Guidelines recommend that all patients with IBD, regardless of their treatment, undergo screening for melanoma given their intrinsic increased risk for melanoma. The Crohn’s and Colitis Foundation preventive guidelines suggest anyone on systemic immunosuppression (azathioprine, 6-mercaptopurine, methotrexate, anti-TNFs, anti-IL-12/23) undergo annual skin cancer screening [53,54]. The added value should be sought in efforts of education, sun exposure prevention, and motivation for screening. Teledermatology might offer a solution to pragmatically include skin cancer screenings into an IBD clinic.

## 8. Lupus-Like Syndrome

Anti-TNF-α inhibitors are well known to induce autoantibodies including ANA (20–60%), anti-dsDNA (15–20%), anti-histones (15–20%), and antiphospholipid antibodies (7–11%) (AE type γ, autoimmunity through immunodeviation). Fortunately, it mainly concerns IgM (instead of IgG) antibodies with low affinity. A correlation exists between ADAs and autoantibodies. However, anti-TNFα-induced lupus is rare and systematic monitoring of autoantibodies without clinical signs should be avoided [55]. In patients with CD and UC, the rate of anti-TNF induced lupus is 0.19–5.7% for infliximab and 0.1–0.6% for adalimumab [17,18]. Start at an older age seems linked to a higher risk [17]. The diagnosis of anti-TNF-α-induced lupus is made by the temporal relation between the symptoms and start of anti-TNF-α therapy and resolution after stop, the presence of autoantibodies (positive ANA or anti-dsDNA), and at least 1 additional sign such as joint pain, fever, fatigue, serositis, or lupus-like skin lesions. The clinical presentation is different compared to classic lupus with generalized non-specific symptoms, skin signs (ulcers, photosensitivity), and less systemic involvement. Compared to other drugs that induce lupus more anti-dsDNA antibodies, more hypocomplementemia, and less anti-histone antibodies are found. Lupus develops after a median of three drug doses and resolves after (a median of) 7 weeks stop of the biologic [56].

Ustekinumab-induced lupus is rare although new onset of lupus and exacerbation of lupus symptoms have been reported [57,58]. Nonetheless, ustekinumab is investigated in clinical trials as a treatment for lupus and can be a good option in IBD patients with coexisting lupus [59]. The data on vedolizumab- and natalizumab-induced lupus are mostly limited to case reports [60,61]. In a summary of 4 years of global post-marketing data of vedolizumab in UC and CD which included 32,752 patients, 17 lupus-related adverse events were reported. Ten of these patients had a treatment history of anti-TNF-α therapy [62].

### Management

Anti-TNF-α induced lupus usually disappears after stopping the offending drug. Topical or systemic corticoids are most frequently used [56]. Antimalarials can be considered in case symptoms and remain present after cessation of the eliciting biologic.

## 9. Vasculitis (Incl Erythema Elevatum Diutinum)

This type of side effect is mainly mentioned here to trigger gastro-enterologists to refer when ‘bizarre’ raised purpuric lesions arise on the skin. Leukocytoclastic vasculitis, usually characterized by palpable purpuric plaques on the lower legs, can be a rare symptom in IBD. It is also reported in about 0.5% of IBD patients treated with TNF-α inhibitors (AE type γ, autoimmunity by immunomodulation) [63,64].

Erythema elevatum diutinum is a rare form of this chronic vasculitis, presenting as red to purple, yellowish to brown nodules or nodi, mainly on acral and peri-articular locations of the body (Figure 1d). Extensor sites of elbows, knees, ankles, and hands are often involved, bilaterally. The skin lesions are asymptomatic to slightly burning or stinging. It can be accompanied by arthralgias, fever, and uveitis/scleritis. It can occur in association with IBD, but also infections, hematologic disorders, and other inflammatory diseases such as arthritis, and celiac disease. Interestingly, as anti-TNF-α therapy can trigger vasculitis, these agents should be considered as causative in this skin finding.

### Management

The causality of the vasculitis can be challenging to determine although drug-induced leukocytoclastic vasculitis resolves after cessation of the biologic in the majority of cases. Rechallenge with the same TNF inhibitor leads to relapse in 67% of patients and a different TNF inhibitor in 33% of cases [63,64]. In severe cases, corticosteroids or immunosuppressive therapy (mycophenolate mofetil, hydroxychloroquine, methotrexate,…) is necessary until recovery [10]. Local or intralesional steroids, dapsone or surgical excision are effective treatments in case of erythema elevatum.

## 10. Lichenoid Reactions

More than 120 cases with lichenoid skin reactions to anti-TNF-α treatment are described (AE type γ, autoimmunity by immunomodulation) [65]. We discern 3 groups: (1) typical lichen planus with purple-pink flat orthogonal papules; (2) non-specific macular or papular morphology; (3) clinically psoriasis, but with lichenoid histologic characteristics. Despite the different clinical presentations, all types are characterized by a lichenoid interface dermatitis confirming the diagnosis. Lichenoid eruptions tend to improve or resolve completely after the stop of TNF-α blockade. Recurrence can occur after rechallenge or change to another TNF-α-blocker [65]. Pigmented lichenoid drug eruptions and lichen planopilaris, leading to irreversible hair loss have also been described [66,67]. Lichenoid reactions have not been associated with ustekinumab, vedolizumab, or natalizumab.

### Management

Lichenoid reactions can be treated with topical steroids or phototherapy. As it is associated with a good prognosis, stopping anti-TNF-α treatment is usually not required [65].

## 11. Granulomatous Reactions

Skin diseases of the granulomatous type in reaction to anti-TNF-α treatment that we highlight here are granuloma annulare, sarcoidosis, and interstitial granulomatous dermatitis (AE type γ, ‘paradoxal’ skin eruption) [68]. Interestingly, despite most data supporting the use of TNF-α antagonists for the treatment of granulomatous disorders, paradoxically, TNF-α antagonist-induced granulomatous skin (and other organs) reactions have been described. *Granuloma annulare* occurs mainly in its disseminated form: brown, to red-yellow annular lesions with a palpable border, of various size. Drug-induced sarcoidosis resembles classic sarcoidosis, with red papules arising with preference in scar tissue. Interstitial granulomatous dermatitis presents as erythematous annular plaques on the trunk and extremities [69] (Figure 1e). In most cases with reactive granulomatous dermatitis in the literature, anti-TNF-α blockade was stopped, although nonetheless, lesions persist in some patients.

### Management

Perform a skin biopsy to show naked granulomas and rule out infectious granuloma (TB, histoplasmosis). Local steroids are first-line therapy. Stopping of anti-TNF-α treatment if needed [70].

## 12. Alopecia Areata/Totalis and Vitiligo

Vitiligo, which presents as depigmented patches on the skin and alopecia areata characterized by patchy or disseminated hair loss, carries a common pathogenesis based on an IFNγ-dominant response. A few cases of new onset alopecia areata/totalis and vitiligo have been described in patients using anti-TNF-α therapy (AE type γ, autoimmunity by immunomodulation) [71,72]. This side effect typically occurs as of 6 weeks after starting therapy, up to 8 months of treatment [73]. A 10-year population cohort study in patients receiving anti-TNF-α showed a hazard ratio of 1.99 (95% CI: 1.06–3.75) for developing vitiligo, while no significant association was found for alopecia areata. The results were, however, not significant in the subgroup of IBD patients [HR = 2.47 (95% CI: 0.69–8.87) and HR = 0.58 (95% CI: 0.09–3.85) for CD and UC, respectively] [74]. Both cases with new-onset vitiligo and alopecia areata on ustekinumab as successful treatments in patients with these disorders have been reported [75,76,77,78,79].

### Management

Stopping the offending biologic (e.g., anti-TNF-α therapy) should be weighed against the severity of the vitiligo or alopecia areata. Both conditions can be treated with topical steroids. Topical minoxidil in the case of alopecia areata and phototherapy in the case of vitiligo can be considered. Switching the class of biologic leads to improvement or at least stop of progression in the majority of cases [71].

## 13. Erythema Multiforme–Stevens Johnson Syndrome–Toxic Epidermal Necrolysis

Cases of these particular skin eruptions that belong to one spectrum have been described (AE type β, hypersensitivity). Erythema exsudativum multiforme presents as erythematous, round maculopapules with a darker center. The typical targetoid lesions primarily occur on palms and soles, and less so on mucosal areas. In Stevens-Johnson syndrome, the eruption is more generalized, including mucosae of mouth and eyes. The skin has the tendency to blister, in its most severe form resulting in toxic epidermal necrolysis (TEN) with a high mortality rate. Fortunately, these syndromes have only been rarely reported with TNF-blockers [80,81]. Based on mostly small case series, there is even evidence that TNF-α-blockade can be useful in these conditions although some controversy remains [82]. Erythema multiforme and Stevens-Johnson syndrome can also be triggered by viral infections such as herpes or other drugs which should also be considered as the underlying cause in patients with IBD receiving TNF-blockers [83].

### Management

Topical and oral steroids can be initiated according to severity. Stopping anti-TNF-α therapy should be considered in case of severe Stevens-Johnson syndrome and TEN [80,81].

## 14. Dermatomyositis

Dermatomyositis (DM)/polymyositis (PM) is a chronic, idiopathic inflammatory myopathy, potentially life-threatening, that affects individuals of all ages (AE type γ, autoimmunity by immunomodulation). Patients with ulcerative colitis are at increased risk for DM (hazard ratio: 6.19 (95% CI: 1.77–21.59) [84]. Even though anti-TNF-α therapy has been cited as a second- or third-line therapy for patients with e.g., refractory juvenile DM, disease flares are possible in DM/PM patients receiving TNF inhibitors [85,86]. There are some reports in the literature regarding the new onset of DM/PM and a specific subset of such diseases such as antisynthetase syndrome in patients affected by other diseases (as Crohn’s disease) during etanercept, infliximab, and adalimumab [87]. In particular, physicians should pay attention to patients with positive antisynthetase antibodies (in particular anti-Jo-1 antibodies) and/or history of interstitial lung disease. In those cases, the use of the TNF-α blocking agents may trigger the onset of PM, DM, and antisynthetase syndrome or may aggravate or trigger the lung disease. A violet-colored or dusky red rash develops, most commonly on the face and eyelids (‘heliotropic rash’) and on the knuckles (‘mechanic’s hands), elbows, knees, chest, and back. Systemic signs such as muscle weakness, fatigue, fever, polyarthralgia, difficulties swallowing, dysphonia are also prevalent. Ustekinumab does not appear to be linked to the development of dermatomyositis and some cases of successful treatment of DM following ustekinumab have been seen [88,89].

### Management

TNF-α blockade should be stopped in patients developing DM/PM while receiving this treatment. In contrast, biologic-naïve IBD patients having dermatomyositis might benefit from TNF-α blockers. First of all, blood analysis of dermatomyositis-specific antibodies and screening for underlying malignancies should be done. The first-line treatment remains high-dose glucocorticoids for at least 4–6 weeks until the dosage can be tapered. Corticoid-sparing agents such as methotrexate, mycophenolate mofetil, azathioprine, tacrolimus, cyclosporine, and cyclophosphamide are useful, especially in patients at high risk for corticoid-induced adverse events such as diabetes, osteoporosis, hypertension, and obesity. In steroid-refractory disease, rituximab is the treatment of choice given its encouraging results in DM/PM [85].

## 15. Diagnosis

Cutaneous side effects of biologic targeted therapies are often diagnosed clinically, especially in cases with clear-cut relationship between the drug and the clinical manifestations or by regression of symptoms after stopping the treatment. An algorithm for clinical diagnostic decision making is proposed in Figure 2.

In case of doubt, a biopsy is warranted. Histopathological diagnosis of these cutaneous side effects is often difficult because of the variety of histopathological patterns and the overlap between patterns that can be seen. A psoriasiform, spongiotic, vacuolar interface, lichenoid, granulomatous, alopecia-like, leucocytoclastic vasculitis, panniculitis, folliculitis, and sclerosing pattern have all been observed.

The development of psoriasiform eruptions has been widely described in patients on TNF-α-inhibitors and the psoriasiform pattern can be regarded as the major histopathological pattern (80%). Clinical psoriasiform lesions have been shown to demonstrate psoriasiform, spongiotic, vacuolar interface, and lichenoid histology or overlap between different patterns. From a histopathological point of view, it is hard to distinguish true psoriasis from a TNFα-induced psoriasiform dermatitis. In addition, up to 35% of psoriasiform biopsies in anti-TNF-α treated patients show a spongiotic dermatitis. Helpful clues in light microscopic diagnosis of TNF-α-induced psoriasiform dermatitis are at least 3 dermal eosinophils per histologic section, the presence of plasma cells, neutrophils in the epidermis (subcorneal), and absence of parakeratosis. Neutrophils in the stratum corneum and papillary plate thinning are clues to idiopathic psoriasis [90,91,92].

The vacuolar interface pattern is seen in dermatomyositis, lupus, and erythema multiforme. A lichen planus-like histology with parakeratosis, deeper infiltrate, and a variable amount of eosinophils and plasma cells should alert the dermatopathologist to a drug-related lichen planus. Interstitial granulomatous dermatitis is the main granulomatous pattern and septal as well as lobular panniculitis have been described as drug-induced panniculitis due to biologic treatment.

In conclusion, in the case of clinical doubt concerning the relationship between the cutaneous eruption and the biologic treatment, a biopsy with recognition of the patterns and clues described above can aid in the diagnosis [93]. This is also summarized in Figure 1. 

## 16. Conclusions

A wide range of cutaneous manifestations is possible in IBD patients receiving biologics. Early recognition is important to adapt the therapeutic approach. While in some cases (e.g., lichenoid reactions), the biologic can likely be sustained, in other conditions (e.g., lupus, dermatomyositis), another therapeutic option should be considered. Cutaneous adverse events are more prevalent with TNF blockers than with ustekinumab. Anti-integrin receptor blockers have the lowest rate or cutaneous eruptions given their gut-selective working mechanism. Although the data are limited, most recalcitrant TNF-induced skin disorders can be adequately managed by switching to ustekinumab or an anti-integrin receptor blocker. A limitation of this review was the non-systematic approach introducing possible selection bias of included articles.

## Figures and Tables

**Figure 1 jcm-10-01040-f001:**
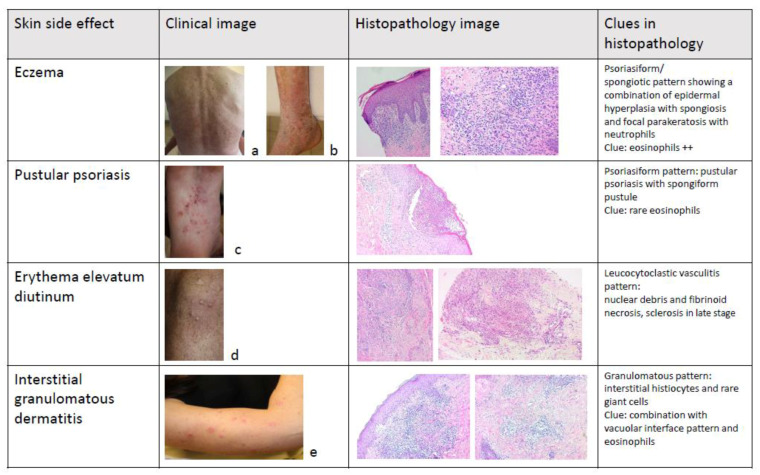
Summary of some characteristic clinical and histopathologic images of skin adverse reactions to biological treatment in IBD (Inflammatory bowel disease) patients.

**Figure 2 jcm-10-01040-f002:**
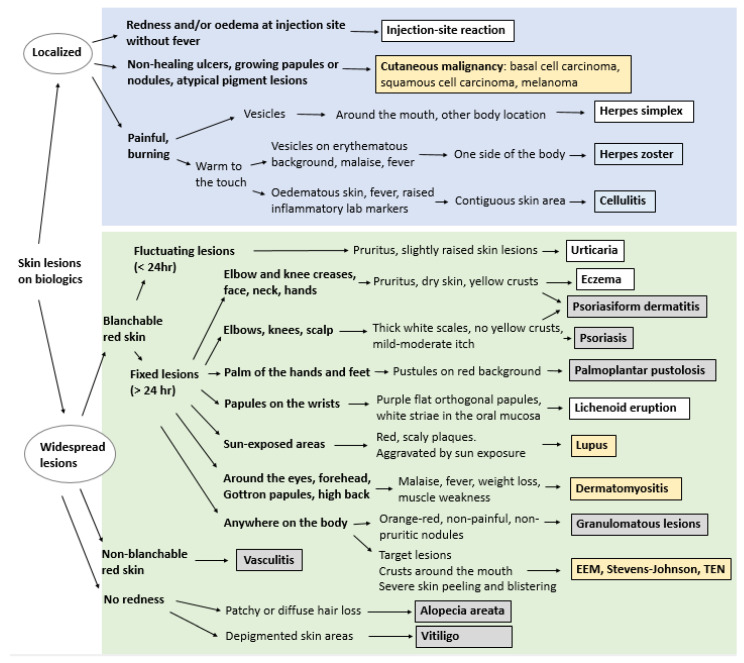
Flowchart illustrating the clinical differential diagnosis of biologic-induced skin lesions for non-dermatologists. *White boxes*: usually no stop or switch necessary; *grey boxes*: consider stop of switch in case of recalcitrant symptoms; *blue boxes*: temporary stop recommended; *yellow boxes*: stop or switch should be considered (for skin cancer: excluding basal cell carcinoma).

**Table 1 jcm-10-01040-t001:** Anti-TNF-α induced skin diseases.

Skin-Related Adverse Events on An-ti-TNF-α (Adalimumab, Certolizumab, Infliximab)	Incidence	Strength of Association
1. Infusion reactions and injection site reactions	<5% and 10%, respectively [11]	Very strong
2. Xerosis and eczema	3–9% [12]	Moderate
3. Cutaneous infections	5–11% [12]	Strong
4. Psoriasiform reactions	4–7% [12]	Strong
5. Cutaneous malignancies	0.3–1.4% [13,14,15]	Mild
6. Lupus-like syndrome	<1% [16]	Strong
7. Vasculitis	<0.5% [17,18]	Strong
8. Lichenoid drug reaction	Cases	Moderate
9. Granulomatous reactions	Cases	Moderate
10. Alopecia areata/totalis and/or vitiligo	Cases	Moderate
11. Erythema multiforme, Stevens-Johnson syndrome, toxic epidermal necrolysis	Cases	Strong
12. Dermatomyositis	Cases	Strong

## Data Availability

Not applicable.

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
