# Peer review of "Cutaneous Manifestations in Biological-Treated Inflammatory Bowel Disease Patients: A Narrative Review"

_jcm, 2021, doi:10.3390/jcm10051040_

Round 1
Reviewer 1 Report
Dears authors;
I am satified about revised version, it is suitable to be accepted now.Â
Kind regards.
Author Response
Thank you for your positive response,
Sincerely,
Jo Lambert
Reviewer 2 Report
As the revision all the comments I suggested are embraced. I still cannot see in the new version Figure 1 and Figure 2. The ratio between diffrent forms of IBD are not equally presented in all cases. Sometimes as % of IBD sometimes in numbers, sometimes just present in UC and CD, there are also some other forms of IBD presented probalby and should be mentioned, which all together sum up till 100% of IBD, % of IBD would be the most infromative.Â
Author Response
Dear Reviewer,
Thank you for your positive response.
The two figures remained the same, and we will send them to the editor. This response system does not allow .png to be uploaded.Â
As for the more precise numbers and percentages, we are afraid that no such data are available. This work summarizes work from others, and does not describe own registry data. What we describe, is - to our knowledge - the best available insight there is.
Nevertheless, we hope that the classification of the side effects, as well as how to handle them makes sense to IBD specialists.
Kind regards,
Jo Lambert, also on behalf of the two co-authors
Â

This manuscript is a resubmission of an earlier submission. The following is a list of the peer review reports and author responses from that submission.
Round 1
Reviewer 1 Report
Paper has a lot of practical clinical implications and may serve as a guideline to the gastroenterologists.
I have minor remarks:
- Title should suggest, that there is a review
- In the introduction please ad the info abour the prevalence of extraintestinal skin manifestations in IBD
- Please explain all the abbreviations, eg. TNF in the line 37, it is explained somehow in the line 41; it should be always when the abbreviations is udes first time
- What about paediatric population? Is the review focused only on adults? If yes plese mark it in the title.
Reviewer 2 Report
Dear authors and editor,
The manuscript titled "Cutaneous manifestations in biological-treated inflammatory bowel disease patients". This is a narrative review of the skin manifestations derived from biological treatment in inflammatory bowel disease (treatment induced-manifestations).
There are many minor and major issues I'd like the authors resolve before consider the paper ready for being published in Journal of Clinical Medicine.
Title:
1-The indication ‘literature or narrative review’ or ‘review of theliterature’ is helpful in clarifying the research design. It is recommended to add
Abstract:
2-Change the keywords. Delete the words "anti-TNFα treatment" and "skin side effect". Not found in the MeSH (Medical Subject Headings). For example: Skin Manifestations ,Tumor Necrosis Factor-alpha, Biological Therapy. Â
3- There are parts of the manuscript that are not referenced:
For example
A) "Management :Lichenoid reactions can be treated with topical steroids or phototherapy. As it is associated with a good prognosis, stopping anti-TNF-α treatment usually not required" Line: 258-260.
B)"Management : Perform a skin biopsy to show naked granulomas and rule out infectious granuloma (TB,histoplasmosis). Local steroids are first-line therapy. Stopping of anti-TNF-α treatment if needed." Line: 273-275
C) "Management : Stop of the offending biologic (e.g. anti-TNF-α therapy) should be weighed against the severity of the vitiligo or alopecia areata. Both conditions can be treated with topical steroids. Topical minoxidil in the
289 case of alopecia areata and phototherapy in the case of vitiligo can be considered. Switching the class of biologic leads to improvement or at least stop of progression in the majority of cases." Line: 286-290.
D) "Management : Topical and oral steroids can be initiated according to severity. Stop of anti-TNFα therapy should be considered in case of severe Stevens-Johnson syndrome and TEN". Line 302-304
4- I recommend including a limitations section. Narrative reviews have a number of design limitations. Also, I recommend a small section on methods to reduce biases. As an example, a literature search was performedfor the present study on the lines of searches foran non-systematic or narrative review, but including features of systematic review methodology. The electronic search included three data-bases, PubMed, EMBASE and Google Scholar, andused three search terms:‘ inflammatory bowel disease’,‘Skin Manifestations’, and ‘Tumor Necrosis Factor-alpha’. The inclusion criteria were: all types of articles, articles publishedin PubMed, and related only to humans. The exclusion criteria were: articles for which full text was not available, were not in English, or were grey literature.
5-Include the references in table 1.
References
- adequate
Reviewer 3 Report
In the era of biologics intestinal bowel disease such as Chron`s disease and ulcerative colitis gain new ways of treatments such as biologics that are targeting TNF-α, IL-12/IL-13 and integrin receptors. However these therapies do carry some risk. 20-25% of anti-TNF-α-treated patients have skin side effects in the course of their treatment. Some cases can be treated topically while others require discontinues of the therapy or switch of the biological therapy. This review represent most common skin problems under biological treatment in IBD patients and could contribute to better diagnosis and proper management of IBD patients.
381-typo mistake; Â Interestingly, Another
410- The Figure 1 is missing
414- The Figure 2 is missing
113- typo mistake INF- TNFα or IFNγ was meant
207- Typo mistake TNFi.and? it was meant TNFα probably?
235- Figure 1 is missing
208- In the table where are written incidence to anti TNFα drugs would be good to put in the bracket the names of the drugs
It would be good if there would be some additional information about the stage of Intestinal bowel disease in each patient that receive the biologics. Some cases are explained more in details while the others lack some explanations. Information such as the type of IBD; ulcerative colitis or Chron`s disease etc., all this would give some additional value to the review and also help later selecting the diagnostic approach. Also would be good if the 5 main categories (α-ε) explained in pathogenies insight under chapter 14 would be at least partially address to each individual skin pathology explained in chapter 1-12.